# M6-10T: A Sharing-Delinking Paradigm for Efficient Multi-Trillion Parameter Pretraining

## Abstract

Recent expeditious developments in deep learning algorithms, distributed training, and even hardware design for large models have enabled training extreme-scale models, say GPT-3 and Switch Transformer possessing hundreds of billions or even trillions of parameters. However, under limited resources, extreme-scale model training that requires enormous amounts of computes and memory footprint suffers from frustratingly low efficiency in model convergence. In this paper, we propose a simple training strategy called "Pseudo-to-Real" for high-memory-footprint-required large models. Pseudo-to-Real is compatible with large models with architecture of sequential layers. We demonstrate a practice of pretraining unprecedented 10-trillion-parameter model, an order of magnitude larger than the state-of-the-art, on solely 512 GPUs within 10 days. Besides demonstrating the application of Pseudo-to-Real, we also provide a technique, Granular CPU offloading, to manage CPU memory for training large model and maintain high GPU utilities. Fast training of extreme-scale models on a decent amount of resources can bring much smaller carbon footprint and contribute to greener AI.

## 1 Introduction

Foundation models with self-supervised learning on big data have become an emerging paradigm of artificial intelligence systems [2], as they mostly possess high transferability to a wide range of downstream tasks and even multiple modalities. The scale of foundation models across domains, including natural language processing, computer vision, and cross-modality representation learning, have been growing tremendously from millions to trillions of parameters [7; 23; 24; 26; 33; 37; 3; 29; 14; 11; 16; 47; 46; 42; 44; 38] thanks to the concurrent advancement in distributed training framework [20; 27; 31; 28; 1; 12; 30] and hardware design, and these studies have made a demonstration of the neural scaling law [13]. However, the training of these transformer-based models incurs high financial costs and even environmental damage due to the massive carbon footprint and thus training extreme-scale models under a decent amount of resources but with high efficiency should be a fundamental goal for both the research and industrial communities to achieve, which promotes the progress of greener AI [22; 34].

Generally there are two tracks of research in large-scale pretraining, dense models and sparse expert models respectively. A typical case of large-scale dense models is GPT-3 [3], a 175-billion-parameter transformer model trained with $10,000$ GPUs for months, incurring striking financial and environmental costs. Researchers have been searching for methods to training large-scale models with a decent amount of costs. Solutions include effective management of memory with gradient and optimizer state partitioning [27] or more efficient model parallelism and pipeline parallelism [37; 20; 12]. A series of following studies apply those techniques to realize fast training of 10-billion-parameter transformers with hundreds of GPUs in $1 - 2$ months. [16; 46; 44; 38] Sparse expert models with large model capacity are capable of fast training owing to the combination of data parallelism and expert parallelism [35; 14; 11; 32], and it is even accessible to train a 1-trillion-parameter transformer with no more than 500 GPUs [42].

Be there as it may, a question emerges in our mind: is it possible to train an extreme-scale model with only a decent amount of resources, e.g., training a 10-trillion-parameter model with around 500 GPUs? Tackling the problem requires the utilization of external memory except for GPU memory, for instance, CPU memory or even NVMe storage [31; 28]. These methods resolve the problem of

high memory footprint, but instead, their extra cost is low training efficiency caused by the frequent swap in-and-out between memories.

In this paper, we provide a solution to training large models that require high memory footprint, and we demonstrate a successful practice of pretraining an unprecedented extreme-scale model with over 10 trillion parameters, an order of magnitude larger than the previous state-of-the-arts [11; 42]. The whole pretraining was conducted on solely 512 NVIDIA-V100 GPUs and lasted around 10 days. A simple and effective training strategy called "Pseudo-to-Real" enables the sharing and delinking of parameters. This training strategy is compatible with architectures built by stacking layers of an identical structure, including dense models like GPT [23; 24; 3], BERT [7], or sparse expert models like M6 [16; 42]. It is essentially a two-stage training. It first trains a relatively small model but with a computation graph of a large one with the utilization of cross-layer parameter sharing, and we name it "Pseudo Giant". Then it builds a correspondingly large model and delinks the parameters of the shared layer for second-stage model initialization. In this way, we achieve fast convergence in the first stage as the training costs much less memory and speeds up with large batches. Parameter sharing that addresses the communication overhead improves training speed as well. The second-stage training is responsible for the final convergence for better performance.

We unlock the secret of pretraining an unprecedented extreme-scale model with over 10 trillion parameters on limited resources of 512 GPUs. Compared with the previous M6-T on around 500 GPUs, we do not have a significant increase in computation resources but level up the model scale by an order of magnitude. Besides the application of the "Pseudo-to-Real" training strategy, we provide a faster offloading mechanism for both management of CPU memory for parameter storage and utility of GPUs. We successfully train the M6-10T within 10 days to reach strong performance in log perplexity evaluation and outperform the baseline M6-T.

Contributions at a glance are below:

- We illustrate the training difficulty of extreme-scale models on limited resources and provide a simple but effective solution called "Pseudo-to-Real". Upstream and downstream evaluation demonstrates the effectiveness of the strategy.
- We further demonstrate a successful practice of pretraining a 10-trillion-parameter model on 512 GPUs and reach an outstanding performance within 10 days.

## 2 RELATED WORK

**Large-Scale Pretraining** In recent years, pretrained language models with growing magnitudes of parameters have been proposed, keeping to raise the validated upper limit of scaling law for model capacity w.r.t the number of parameters [13]. Earlier milestones of extreme-large models come from GPT-2 [24] and Megatron-LM [37], which demonstrates that scaling the transformer model up to billions of parameters can result in improvement on language modeling benchmarks [19; 21]. Turing-NLG [33], as a successor, implements a 17-billion-parameter transformer and achieves further lower perplexity. Similar phenomena are also observed on classification language tasks by T5 model [26]. The GPT-3 [3] pushes the boundary of model scale to 1,750 billion parameters and demonstrates its striking effectiveness on downstream tasks in even zero-shot settings. Furthermore, large-scale pretraining has recently demonstrated success in other fields, including pretraining on other languages or cross-lingual pretraining [41; 46; 44; 6], cross-modal pretraining [25; 29; 16; 8; 48] and code generation [4]. Along with the benefit from increasing the model scale, the concern of unaffordable pretraining cost in time, computation resource and energy keeps emerging [22; 2], resulting in the strong demand for more efficient and greener large-scale pretraining [34].

**Methods to Train Larger Models Faster** Our work is about designing algorithm to train extreme-scale models efficiently. Researchers have been demonstrating different types of methods to reach this objective. To reduce the computational cost during training, some studies introduce sparsity to the model. Mixture-of-Experts (MoE) [35] was proposed and was reintroduced in Mesh Tensorflow [36]. It shows that MoE with sparsity can significantly improve the model scale efficiently without increasing computation, and the models achieved state-of-the-art performance in language modeling and machine translation. GShard [14] extended it to a tremendously large scale of 600B parameters with the sophisticated collaborated design of model architecture and demonstrated its effectiveness

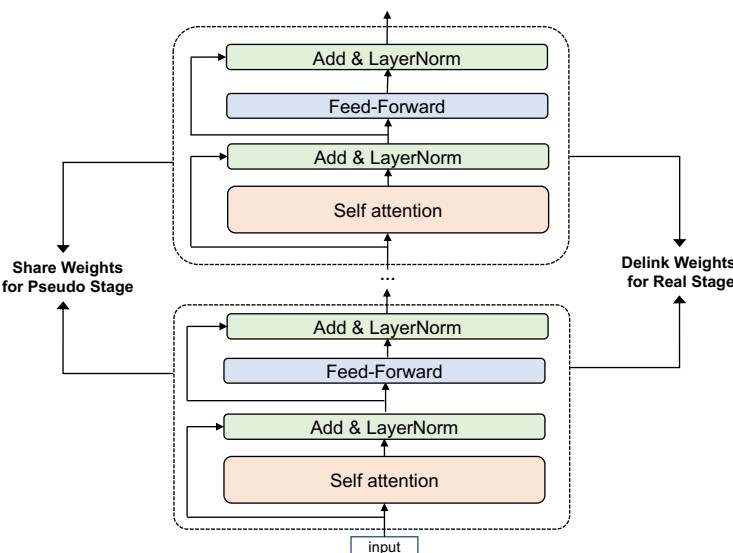

Figure 1: A demonstration of the Pseudo-to-Real training paradigm. It first shares parameters across layers at the Pseudo stage, and then delinks the parameters at the Real stage.

across over 100 languages. Similarly, Switch Transformer reached 1.6 trillion parameters and showed its strong performance on NLU tasks. Those models with high sparsity are computationally efficient, and therefore though they possess large model capacity they still can be trained with high efficiency. Most other studies still focus on training dense models to validate the scaling law, and thus the emerged problem is the distributed training of large models. The most influential distributed framework should be DeepSpeed that proposed ZERO [27] that partitions optimizer states and gradients to multiple GPU devices, and ZERO-offload [31] as well as ZERO-Infinity [28] can even offload parameters to CPU memory and NVMe storage. The memory footprint management makes training extremely large models on limited resources possible. However, such offloading mechanisms still have some defects that they may fail to fully utilize the fast hardware. For example, when offloading all parameters to the CPU, the GPU memory can be idle. In this work, we tackle this issue by proposing a granular offloading mechanism that can determine which parameters to be offloaded.

In addition to reducing the amount of computation in a single iteration, another route to speedup training is to reduce the needed iterations for model convergence. Child et al. [5] and Xiong et al. [40] propose to put forward the layernorm operations in transformer blocks for more stabilized and faster convergence. You et al. [43] employs a layer-wise adaptive optimizer to enable super-large pretraining batches. Zhang & He [45] progressively increases the layer dropping rate in a stochastic manner which significantly speedups pretraining.

## 3    APPROACH

### 3.1    MODEL ARCHITECTURE

Choice in model architecture depends on several factors. First, the architecture should contain a sequence of stacking layers, as the sequential structure enables cross-layer parameter sharing. We prefer a simple encoder or decoder architecture, instead of an encoder-decoder framework where there are cross attentions that bring extra parameters and incur difficulties in activation checkpointing. Second, a model of such architecture should be compatible with different types of downstream tasks, including understanding and generation, and it is more preferable that it can be compatible with multiple modalities. Third, as we mention that dense models and sparse expert models are two main tracks of large-scale pretraining, we prefer the model that can flexibly become whether dense or sparse expert models. Therefore, we select M6 [16] as an option, and we evaluate the effects of our method on M6 of different scales and types.

M6 is built with stacking transformer layers, which includes self attention and feed-forward neural nets (FFN). For the transformation from dense models to sparse expert models, we should only replace FFN layers with the Mixture-of-Expert (MoE) layers. MoE consists of multiple experts, which are usually FFNs distributed on different devices. A gating network decides the dispatching and combining behaviors of each token and thus tokens can be processed in diverse devices. Such mechanism is a combination of data parallelism and expert parallelism, and thus it is highly efficient though with large model capacity. For the training, to realize the learning of both understanding and generation, the model is trained with text denoising and language modeling on plain text data and with image-based text denoising and image captioning on multimodal data. The model is compatible with different types of downstream tasks and can process information of multiple modalities.

## 3.2 PSEUDO-TO-REAL

This section demonstrates the details of "Pseudo-to-Real" two-stage training strategy that enables fast training of high-memory-footprint-required transformer models. The strategy consists of two stages. The first stage trains a model with many fewer parameters but with a large computation graph ("Pseudo Giant"), and the second stage trains a corresponding large model ("Real Giant") initialized with the delinked weights of the shared layer. Thus we name the strategy "Pseudo-to-Real", and the general idea is demonstrated in Figure 1.

### 3.2.1 "PSEUDO" STAGE: LAYER-WISE PARAMETER SHARING

The core of "Pseudo" stage is to train a Pseudo-Giant that shares parameters across layers. Cross-layer parameter sharing has proved successful in maintaining satisfactory performance while keeping a much smaller amount of parameters. We introduce it to training an extreme-scale model, and we hypothesize that the first-stage training can gain benefits from cross-parameter sharing as it can address communication overhead and it consumes much less memory footprint. Also, as Pseudo Giant with much fewer parameters is not bounded by memory, it can be trained with large batches for acceleration.

Suppose we build an M6 model with $L$ layers that share parameters across all layers. The Pseudo Giant though consists of a computation graph of a $L$-layer transformer, its number of weight parameters and optimizer states should be $1/L$ of those of the original one. As to the gradients, we can accumulate the gradients of each layer in the backward computation process, and therefore the amount of gradients becomes $2/L$ of the original one. Such saving in memory enables much faster training with larger batches. Also, it is capable to use fewer resources even due to lower memory consumption.

This can also be applied to sparse expert models, as their architecture is stacking transformer layers. However, different from dense models, the MoE partition their weights to all devices, and each token is distributed to a selected expert by a routing function. Thus the models can trained with data parallelism across all devices and expert parallelism where parameters at each device are activated. This mechanism though enlarges model capacity significantly without much efficiency loss, yet it limits the flexible usage of GPU resources in different stages, say training and inference. Due to the application of data parallelism, we can implement dense models on different numbers of GPU devices at the training and inference stage. However, as mentioned above, the combination of data parallelism and expert parallelism requires the identical amount of devices at different stages.

To tackle this problem, we specifically design methods for expert merging and partitioning. Due to cross-layer parameter sharing, at the Pseudo stage it is capable of implementing more experts on each device, and it is available to use fewer GPUs while keeping the total number of experts. We name it expert merging for simplicity. At the Real stage, the delinking of parameters with the same amount o devices will cause out-of-memory errors. We delink the parameters and partition experts to more devices, e.g., from $256$ to $512$, and thus there are fewer experts on each device where memory is sufficient. Thus we name it expert partitioning. Therefore, now it is available to use different number of GPU devices at different stages for training sparse expert models.

### 3.2.2 "REAL" STAGE: DELINKING THE SHARED PARAMETERS

We name a large model without cross-layer parameter sharing "Real Giant", in comparison with Pseudo Giant. Both models share an identical computation graph, but possess different numbers

Table 1: Experimental results on downstream evaluation of natural language understanding. We evaluate the performance of models on 8 tasks of GLUE dev set except for WNLI following Devlin et al. [7].

| Model | #Params | SST-2 | CoLA | MNLI | QNLI | QQP | MRPC | RTE | STS-B | Avg |
|-------|---------|-------|------|------|------|-----|------|-----|-------|-----|
| BERT | 345M | 93.7 | 60.6 | 86.6 | 92.3 | 91.3 | 88.0 | 70.4 | 90.0 | 84.1 |
| M6 | 350M | 94.2 | 61.3 | 86.3 | 92.3 | 91.4 | 91.4 | 83.4 | 89.0 | 86.2 |
| M6 (P) | 65M | 90.5 | 46.2 | 80.8 | 89.4 | 89.9 | 90.1 | 70.7 | 85.6 | 80.4 |
| M6 (P2R) | 65M/350M | 94.4 | 58.6 | 86.6 | 93.3 | 91.5 | 91.5 | 84.5 | 89.1 | 86.2 |

of parameters. Given a Pseudo Giant fully trained until convergence, we apply the delinking of cross-layer shared parameters to accelerate Real Giant training. There is no need to train a large model from scratch. The model can start its convergence from low perplexity.

Embedding initialization can be directly restored, but the layer weights should be treated specially. In practice, there is only one layer of weights $\theta_{shared}$ in Pseudo Giant, and there are $L$ layers of weights $\{\theta_1, \theta_2, \cdots, \theta_L\}$ in Real Giant. Thanks to their identical structure, each layer of Real Giant can be initialized with $\theta_{shared}$. Without further training, this model is equivalent to a fully-trained Pseudo Giant.

This extremely simple training strategy is highly beneficial for the high-memory-footprint-required large models, especially extreme-scale models like the 10-trillion-parameter M6. While the first stage of training saves much time for faster convergence, we can use a decent amount of computational resources in this stage as lower efficiency in this stage becomes acceptable. Therefore, in the practice of training an extreme-scale M6, we apply CPU offloading to utilize CPU memory. Therefore, we can use a limited amount of resources, e.g., 512 GPUs, to train an unprecedented 10-trillion-parameter model efficiently, which is an order of magnitude larger than the state-of-the-arts.

### 3.2.3 TIMING FOR SWITCHING

A question naturally emerges: when should we switch from the Pseudo stage to the Real stage? As mentioned above, the greatest advantage of Pseudo stage for training is the significantly faster convergence speed. Yet the performance of Pseudo Giant is bounded by its limitation in the number of parameters. Training Pseudo Giant until convergence apparently incurs much waste of time.

In practice, we present a simple strategy to determine the training step to switch from Pseudo to Real based on the convergence speed. During the training of the Pseudo stage, we evaluate a training step in a fixed interval by attempting to transfer it into the Real stage and training for a small while (e.g., 30 minutes). After that, we will revert the model parameters to the evaluated step and continue the training of the Pseudo stage for the same training time as the Real stage. If the decreasing speed of loss in the Real stage surpasses that of the Pseudo stage, we determine the evaluated training step as the best switching point for the next-stage training.

### 3.3 EXPERIMENTS

In this section, we demonstrate experiments to evaluate the effectiveness of the training strategy by evaluating the model quality and observing the performance improvement.

### 3.3.1 PERFORMANCE EVALUATION

We aim to validate two hypotheses: 1. Pseudo-to-Real training paradigm can help the model reach competitive performance with the one trained from scratch without parameter sharing; 2. Pseudo-to-Real can effectively accelerate convergence in training of large-scale models.

We aim at discovering the model quality brought by different pretraining strategies. "Pseudo (P)" refers to training with parameter sharing across layers, and "Pseudo-to-Real (P2R)" refers to the proposed two-stage training of a sharing-delinking paradigm. We pretrain the model on BookCorpus [49] and English Wikipedia [7], which are corpora with around 16GB of plain texts. Following Radford et al. [23] and Lewis et al. [15], we use a vocabulary of around $50,000$ subwords. Each

Table 2: Experimental results on downstream task evaluation. "#Params" refers to the number of parameters. We report the PPL evaluation on WikiText-103 and the ROUGE1, ROUGE-2, and ROUGE-L evaluation on Gigaword.

| Model | #Params | WikiText-103 | Gigaword |
|---|---|---|---|
| Megatron-LM | 350M | 16.69 | - |
| UniLM | 340M | - | 38.5/19.5/35.4 |
| M6 | 350M | 16.59 | 38.8/20.1/36.0 |
| M6 (P) | 65M | 28.60 | 36.9/18.1/34.3 |
| M6 (P2R) | 65M/350M | 16.60 | 38.3/19.3/35.7 |

Table 3: Model refers to the types of model. $d_{model}$ and $d_{ff}$ refer to the hidden size and intermediate size. $L$ refers to the number of layers in the computation graph, and $l$ refers to the number of transformer layers with parameters. #Heads refers to the number of heads in self attention. #Params refers to the total number of model parameters. We also report their training speed on $48$ GPU devices by the number of consumed samples per second.

| Model | $d_{model}$ | $d_{ff}$ | l/L | #Heads | #Params | Speed |
|---|---|---|---|---|---|---|
| Base | 1024 | 4096 | 24/24 | 16 | 350M | 650 |
| Pseudo | 1024 | 16384 | 1/36 | 16 | 1.4B | 248 |
| Real | 1024 | 16384 | 36/36 | 16 | 1.4B | 48 |

sample consists of sentences from an identical passage, and we use a sequence length of $512$ and correspondingly truncate or pad the sequence. Following the common practice in pretraining [7; 17], we apply AdamW optimizer [18] for optimization. To determine the most suitable learning rate of the two stages for fast convergence, we have made some preliminary tests and finally used the peak learning rate of 2e-4 for "Pseudo" stage and 8e-5 for "Real" stage, respectively. We use a cosine decaying mechanism for learning rate scheduling and a warm-up ratio of $0.001$. We pretrain the models until convergence and transfer them to downstream tasks.

For comprehensive analysis, the experiments include natural language understanding and generation tasks. Notably, for NLU tasks, we follow Devlin et al. [7] and validate transfer effects on GLUE [39]. For natural language generation, we specify zero-shot language modeling and text summarization to evaluate both models' upstream and downstream quality on generation. Specifically, we conduct experiments on WikiText-103 and Gigaword for both tasks.

Experimental results on NLU tasks are demonstrated in Table 1. For better comparison, we also present the experimental results of BERT with a similar amount of parameters, and it shows that M6 can achieve a better performance over the baseline. From Table 1, we find that Pseudo-only training brings a significant performance downgrade in the $8$ tasks. It is fair to believe that the limited amount of parameters hinders downstream performance, though with a large computation graph. However, in comparison, we find that P2R can help the model perform similarly to the one trained from scratch without parameter sharing. Specifically, M6 with P2R can even outperform the baseline on 6 tasks, including SST-2, MNLI, QQP, MRPC, RTE, and STS-B. We further validate their performance on zero-shot language modeling and text sumarization tasks. Table 2 demonstrate the model performance on both tasks, and we additionally add Megatron-LM [37] and [10] for better comparison. Similarly, Pseudo-only training leads to far worse performance, and P2R is also able to help the model achieve similar performance with Real training.

### 3.3.2 EFFICIENCY EVALUATION

Next, we evaluate the efficiency of training strategies by observing model performance under the condition of identical computation budget.

**Experimental Setup**   To satisfy the requirements of high memory footprint where the two-staged training can make a difference in training efficiency, we conduct experiments on a 1.4B-parameter model. We present the details of model configuration in Table 3.

Table 4: Experimental results of large models trained with limited budget on downstream task evaluation. We report the PPL on WikiText-103 and the ROUGE scores on Gigaword.

| Model | #Params | WikiText-103 | Gigaword |
|---|---|---|---|
| **limited budget** | | | |
| M6-1B (P) | 90M | 56.79 | 36.8/17.9/34.1 |
| M6-1B (R) | 1.4B | 26.80 | 36.9/18.2/34.2 |
| M6-1B (P2R) | 90M/1.4B | 23.60 | 37.3/18.3/34.5 |

For both Pseudo and Real Giant, we train them with a total batch size of $6,144$. In practice, we use a micro-batch size of $32$ and a gradient accumulation step of $4$, and we train our models on $48$ NVIDIA-V100 GPU devices. We compare the model performance with a limited budget, where models have been trained for the same duration of time. Correspondingly, "Pseudo" has been trained for around $25,000$ steps, Real has been trained for around $4,700$ steps, and "P2R" has been trained for $12,000$ steps.

**Model Performance**  Table 3 demonstrates the training speed of the models. We report their training speed on $48$ NVIDIA-V100 GPU devices with their consumed samples per second. Training Real Giant model from scratch is highly time-consuming. Pseudo Giant training has an advantage of around $5$ times of convergence speed over Real Giant training. We also observe the loss convergence of both P2R and Real Giant trained from scratch. In Figure 2 we present the development of pretraining language modeling loss, which is the log perplexity, on the time basis. The log perplexity of P2R decreases much faster than that of the Real Giant, with an advantage larger than $0.3$.

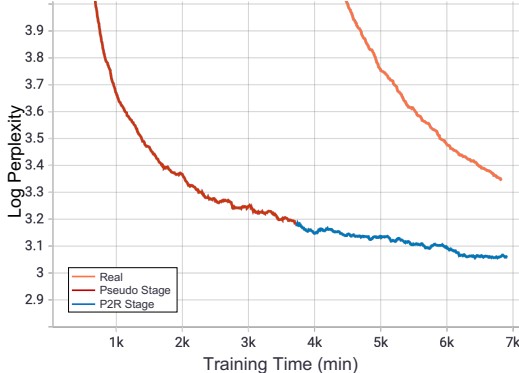

Figure 2: Comparison of pretraining language modeling loss of M6-1B P2R and Real on time-basis.

Experimental results on downstream tasks presented in Table 4 are consistent with our hypothesis that Pseudo-to-Real training can speed up training effectively. In the setup of limited budget, the M6 model trained with Pseudo-to-Real can outperform the Pseudo Giant and Real Giant in both language modeling and text generation.

## 4  TOWARDS A 10-TRILLION-PARAMETER MODEL

Previously, training large-scale models brings tons of challenges to the collaboration algorithm design, distributed training, as well as hardware design, etc. Training a GPT-3 of $175B$ parameters with a combination of data parallelism on over $500GB$ of data should cost around $300$ GPU-years. Later with the emergence of partitioning on optimizer states, gradients, and even weights, GPU memory can be fully utilized without performance degradation. Now we can even use the CPU memory or even NVMe storage to store the parameters, but we have to bear the costs of efficiency. Therefore, we attempt to tackle the difficulty of extreme-scale model training from the perspective of algorithm design and thus we apply the aforementioned Pseudo-to-Real training strategy to train an extreme-scale model.

### 4.1  MODEL SETUP

We design a 10-trillion-parameter M6 model with the combination of existing methods and proposed strategies to demonstrate a case of how to train an extreme-scale model efficiently. In comparison with the previous studies of trillion-parameter models [11; 42], this one is almost 10 times larger. To efficiently utilize the memory, we adopt Mixture-of-Experts and we replace every FFN layer with the memory-efficient MoE layers. Notably, we remove the auxiliary loss that consumes memory and

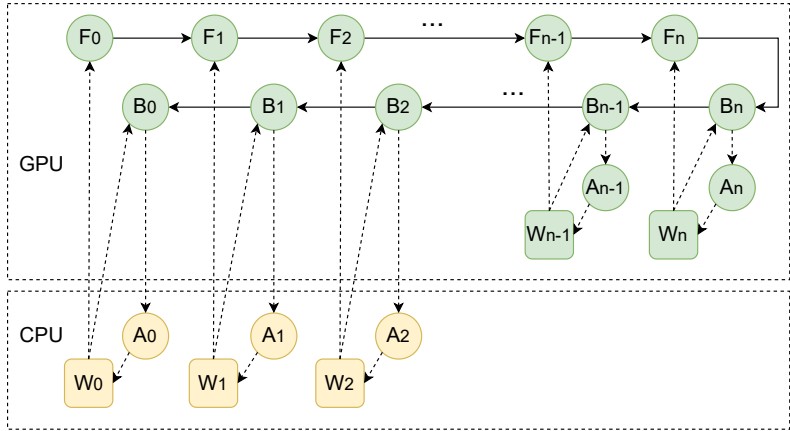

Figure 3: A demonstration of Granular CPU Offloading mechanism.

demonstrates little effects on model quality, and we follow Yang et al. [42] to apply expert prototyping for improved model quality and training stability. To be more specific, the hidden size $d_{model}$ is 1024 and the intermediate size $d_{ff}$ is 9984. The number of model layers is 48. For each MoE layer, there are 10240 experts distributed on multiple devices. We use 80 prototypes of experts based on our experience in preliminary experiments. The training batch size per GPU device is 8. We implement models on EFLOPs, a distributed training platform with an advanced server architecture and a new network topology [9]. Specifically our models are trained on a cluster of 8-GPU workers connected by RDMA networks with a bandwidth of 100Gb. The CPU memory of each worker is around 750GB. The expert distribution and Granular Offloading strategies are supported by Whale framework[12].

## 4.2 GRANULAR CPU OFFLOADING

To utilize CPU memory for larger models with fewer resources, we apply the Granular CPU offloading which has a higher efficiency compared with the conventional CPU offloading. Previously we note that conventional offloading mechanisms offload all parameters, which may fail to effectively utilize GPU. To improve the efficiency of CPU offloading, we propose a new CPU offloading mechanism called Granular CPU offloading. The training process is composed of phases including "Forward (Fn)", "Backward(Bn)" and "Apply (An)". Offloading all model parameters to CPU in Fn and Bn requires loading parameters from CPU to GPU memory twice. Activation checkpointing that brings recomputation needs the parameters loaded in Bn. An requires the gradients offloaded from GPU memory to CPU memory. Assume the model parameter size is $\mathbf{W}$, the above processes bring parameter movement of size $\mathbf{4} * \mathbf{W}$.

In offloading, PCIE is the bottleneck of the whole training process. We observe that when offloading all parameters with recomputation, the GPU memory is idle. We can fill up the GPU memory by selective offloading, leaving part of the model in GPU memory. In this way, the model can be accelerated by reducing across-device memory copy. In our preliminary experiment, with the setting of training a 48-layer 78B-parameter M6 model on 8 NVIDIA-V100 GPU devices, the step-time costs 89 seconds when fully offloading the parameters into CPU memory. In comparison, granularly offloading the first 24 layers into CPU memory and leaving the remaining 24 layers on GPU reduces the step-time to only 45 seconds. The significant difference in training step-time indicates that the time-cost of parameter movement between CPU and GPU will dominate the training step-time when offloading is employed, thus the granularity of offloading and the utilization of GPU should be seriously considered in extreme-scale pretraining. In addition, offloading the whole model can result in OOM error in the CPU when the model is extremely large.

With Granular CPU offloading, we successfully implement a 10-trillion-parameter M6 model on solely 512 NVIDIA-V100 GPUs. Furthermore, at the Pseudo stage, we can train a Pseudo Giant with a computation graph of 10 trillion parameters only with 256 GPU devices without the utilization of CPU memory for offloading. Thus in our practice, we train a Pseudo Giant with only 256 GPUs, and

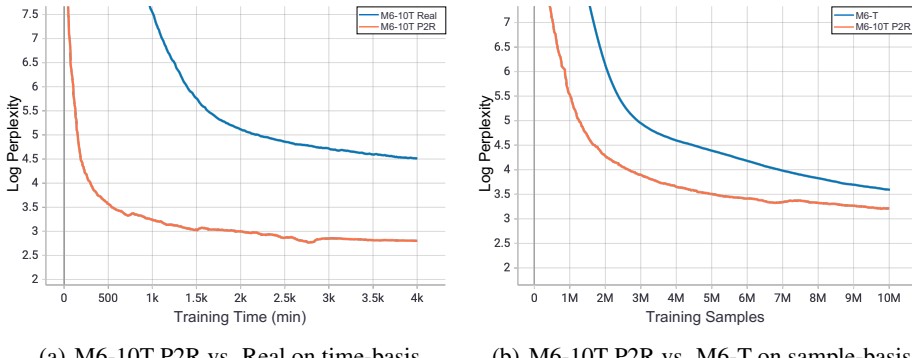

(a) M6-10T P2R vs. Real on time-basis      (b) M6-10T P2R vs. M6-T on sample-basis

Figure 4: The log perplexity of M6-10T Pesudo-to-Real (P2R) compared with baselines on time-basis and sample-basis, respectively. (a) Compared with M6-10T Real on time-basis, M6-10T P2R converges much faster due to the significant reduction of time-cost on offloading in the Pseudo stage. (b) Compared with 1T-parameter M6-T model on sample-basis, though undergoing the Pseudo stage which limits the model capacity, M6-10T still has an advantage in training sample efficiency.

then partition the experts and redistribute them to 512 GPUs. This saves the usage of GPU resources, which is more resource-efficient and also more environmentally friendly.

### 4.3 ANALYSIS

**Training Efficiency** We pretrain the M6-10T with Pseudo-to-Real training strategy for around 10 days, and we additionally train an M6-10T from scratch without the strategy for around 3 days for comparison. The Real stage training can be facilitated without Out-of-Memory errors with the help of Granular CPU offloading, but its step-time is only around 180s. In contrast, the step-time of Pseudo stage is only 14s without the cost of offloading, which greatly boosts the training efficiency of M6-10T P2R. The M6-10T with P2R has been trained for $15k$ steps, but the one from scratch has been trained for solely $1.3k$ steps. We record the log perplexity of both models trained on the M6-Corpus on the time basis in Figure 4(a). Results show that within the same length of time M6-10T with P2R can outperform the one trained from scratch by a large margin.

**Convergence Analysis** We have trained M6-10T with Pseudo-to-Real strategy for around 10 days, and the model converges to a low level of log perplexity based on the upstream evaluation on the M6-Corpus. For better comparison, we also show the convergence performance of the 1T-parameter M6-T model proposed in the previous work. As shown in Figure 4(b), the observation is consistent with our intuition that the model with a larger capacity can converge faster on the sample basis, and it should achieve the best performance on language modeling. What leaves open is whether it can positively lead to better downstream performance concerning different types of downstream tasks. Finetuning extreme-scale models should be difficult and there is still much room for us to discover the potential of extreme-scale models.

## 5 CONCLUSION AND FUTURE WORK

Pseudo-to-Real training strategy is a simple and effective way to train large-scale models that are highly memory consuming, and it is also essential to training extremely large models with limited resources with significantly higher training efficiency. We unlock pretraining unprecedented extreme-scale models with 10 trillion parameters with limited resources of 512 GPUs in 10 days. Besides the application of Pseudo-to-Real training strategy, we further provide Granular CPU offloading to enhance GPU utility while breaking the GPU memory wall with a cost in efficiency. The advances take a leap towards extreme-scale model training beyond implementation on limited resources. With only a few GPU cards, training large models with tens or hundreds of parameters has become accessible to many researchers. We believe that this can motivate low carbon dioxide production and encourages the progress of green AI.

## ETHICS STATEMENT

This work is highly concerned with large-scale language models and multimodal pretrained models. These models have been pretrained on broad data of plain texts and image-text pairs, which might contain harmful information, such as hate speech, terrorism, pornography, etc. We have put much efforts to remove these kinds of data in our datasets by quality evaluation on texts and images. However, this problem cannot be eliminated and ignored, and it is common in the pretraining community. For those models that are not trained on commonly-used public datasets, we will carefully release the model checkpoints before careful evaluation, and also limit the access to avoid misconduct.

## REPRODUCIBILITY STATEMENT

This work is generally reproducible. Following the description in Section 3, researchers can easily implement the training strategy on the codebases for pretraining, including Huggingface Transformer [1], Fairseq [2], etc.

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
