# OpenReview forum: "M6-10T: A Sharing-Delinking Paradigm for Efficient Multi-Trillion Parameter Pretraining"
_ICLR.cc/2022/Conference — ICLR 2022 Submitted_

### Official Review · Reviewer_BFav · 2021-11-02

**Correctness:** 3
**Technical Novelty And Significance:** 2
**Empirical Novelty And Significance:** 2
**Recommendation:** 6
**Confidence:** 4

**Main Review:**

Strengths:

1. The proposal training strategy is novel and interesting, and has the potential to become the standard practice for training transformers.  I personally will certainly try it for my next large language model training.

2. The proposed strategy is simple and straightforward to implement.

3. Experiments are conducted on real systems and on real datasets.

Weakness or areas of improvements.
In general, this paper will be greatly improved with a better experiment design.

1. It's unclear whether the M6-1B model with longer training time in Table 2 was started from P2R strategy or from the Real Run. Actually it's better to list the results from both to study how P2R affects the long term convergence of the model. I'm interested in see what happens to Figure 2 after 7k mins of training time until convergence. Will M6-1B (P2R) perform worse than M6-1B (Real) at convergence?

2. It' unclear to me what speed-accuracy was achieved by the proposed P2R strategy from the experiments section.  There are a lot of unanswered questions such as: what's the budget set in table 2?  How are the downstream task performance (few-shot) changed with the compute budget?  How much quality loss is introduced by the 5x saving in the compute cost during the pseudo phase? I would suggest the authors conduct more thorough studies to help readers understand the pros and cons of P2R.  These additional experiments can be done with little additional training cost as P2R also applied to smaller models as long as the stacking layers are identical in structure.  For example,  I would expect that  a larger model (like Transformer-Big) trained with P2R will have less training time but worse performance than training from scratch (like Transformer-Big Real) , but still outperforms the base model (like Transformer-Base) in model quality with compatible training cost. Something like the table below:

Model &  Training time &.  pplx & one-shot downstream performance.

Transformer-Base (Real) &. 3  & 2.5 &  40\\

Transformer-Big (P2R) &. 3 &  2.2 & 45 \\

Transformer-Big (Real) &. 10  & 2.1 & 50 \\

3. In Table 2, M6-1B's Gigaword performance is worse than M6 with smaller scale, which is unexpected. I disagreed with the author's conclusion that the worse performance is not related to the training strategy.

4. I think the significance of the proposed strategy to large language models is actually limited.  First, it's not going to save the serving cost. Second, the amount of  time saving from the P2R approach might be small compared to the overall convergence time, especially for the 10T scale.  It will be more interesting to see how this approach works with regular size transformer in various applications.



**Summary Of The Paper:**

The paper proposed an interesting strategy to reduce the training time for large scale language models consisting of stacking layers with identical structures. Users first trained the models with shared parameters across the layers, then relax the tie constraints so that parameters at different layers are updated differently.  The paper showed some empirical evidence that the proposed strategy converged faster given a limited training time budget and demonstrated the feasibility of training a 10T model.

**Summary Of The Review:**

The paper proposed an interesting idea to reduce training time for models with identically repeated layers. The empirical evidence provided in the experiments are not convincing enough to make a standard practice.  However, it's still worth a try for practitioners in the model pretraining community.

---

> ### Author Response · Authors · 2021-11-17
> **Authors' response**
>
> Thank you very much for your helpful reviews! For the raised concerns, our explanation is given below:
> 1. For the comparison of performance between P2R and Real models when training longer, we have added a section with corresponding experiments in the updated version of this paper. According to the results, the performance of P2R can reach the Real model on various downstream tasks. Please refer to section 3.3.1 in the updated version for more details.
> 2. For the performance of the Pseudo model without P2R training, we conduct experiments on both the setting of limited budget and longer training (section 3.3 in the updated version), the experimental results indicate that the Pseudo model sacrifices significant downstream performance in return of much faster speed, for which reason a following P2R phase is necessary to overcome this issue.
> 3. For the potential value of pseudo-to-real paradigm, we believe our proposed approach applies to both large (10T-param) and regular model sizes (1B-param) and the experiments are conduct on both settings in the paper. In the future works, we will try to explore the paradigm on more model architectures and downstream tasks.

---

### Official Review · Reviewer_pBTW · 2021-11-04

**Correctness:** 2
**Technical Novelty And Significance:** 3
**Empirical Novelty And Significance:** 2
**Recommendation:** 3
**Confidence:** 5

**Main Review:**

The paper tackles an important problem of improving the efficiency (time and resource) of training massive models, as these models rapidly scale in size to achieve greater performance. P2R seems to be a promising approach that aligns with the intuition that the structure of the model (and data) for effective learning could be dynamic throughout the training process. I believe this intuition is the basis of successful schemes such as pre-training+finetuning, lottery tickets hypothesis, etc.

Two major weakness of the paper are:
1. Does not demonstrate P2R for M6-10T, the 10T parameter model, since no downstream task results are presented for this model in the evaluation section. Similarly, the claim of 10 days pre-training remains unsupported by the draft.
2. The parameter sharing approach seems arbitrary and un-insightful. Why have all the layers share 1 set of parameters, which not 2 or 4, or something fewer than number of layers?

**Summary Of The Paper:**

The paper proposes a technique called Pseudo-to-Real (P2R) for reducing the computational and time requirements of training massive (or Giant) models with trillions of parameters. The key idea of P2R is a two phase training approach for Giant models. The first phase involves training a smaller version of the model (a.k.a., Pseudo-Giant) which is obtained by making all layers share parameters. The second phase involves training the Giant model after initializing with Pseudo-Giant weights. The paper further proposes Granular CPU offloading which is to offload some but not all model parameters to CPU memory to reduce GPU memory consumption. Finally, the paper provides some evaluation results to demonstrate P2G.

**Summary Of The Review:**

Although the problem is important and the proposed P2R approach is promising, the submission does not demonstrate the effectiveness of the proposal on the problem. Specifically, the paper lacks downstream task results of a 10T model trained with P2R. It is difficult to fairly judge this paper without these qualitative results, especially given the claims.

---

> ### Author Response · Authors · 2021-11-17
> **Authors' response**
>
> Thank you for your helpful review! Below we address your concerns:
> 1. About the downstream performance of M6-10T: Due to the limitation of computation resources and time budget, currently we could not provide downstream results and thus we evaluated the model by log perplexity. According to the experience of pretraining on smaller-scale models, in general we believe the pretraining PPL and downstream performance are in positive correlation. In the future, we will attempt to add more downstream experiments for the extreme-scale model.
> 2. About the setting of shared layers: In fact, the number of layers to share is a choice of hyperparameter. As we have Real stage for final convergence, we choose the extreme case for Pseudo so that the time saving can be the most. Since the paradigm is ultimately proposed for effectively training a 10T-param model, only keeping a single shared layer is essential for us to train the model without using CPU offloading which greatly degrades efficiency. In the future, we will cover more experiments on the choice of shared layers to evlauate the trade-off between efficiency and model performance.

---

### Official Review · Reviewer_YjPt · 2021-11-05

**Correctness:** 4
**Technical Novelty And Significance:** 2
**Empirical Novelty And Significance:** 1
**Recommendation:** 3
**Confidence:** 4

**Main Review:**

The proposed approach makes sense. However, I have the following major concerns:

1. The 10-trillion parameter model (M6-10T P2R) might be under-trained. It is hard to tell if training a relatively smaller model (e.g, 100 billion parameter) on the same number of GPU hours can give better numbers. I believe a larger model can outperform smaller models only if it's trained enough. The authors advocate for green AI, and train a 10-trillion parameter model. Thus, the main point should be the 10-trillion parameter model uses fewer GPU hours to get the same quality. I do not see such evidences.
    * In the "Convergence Analysis" paragraph and Figure 4, the authors showed that M6-T converges faster than M6-T in terms of the same number of seen examples. But I think the more important question is, does M6-T achieve better numbers than M6-T given the same GPU hours yet.
    * In Table 2, how long does "training for long" mean? Additionally, if there is only "limited budget", the important question is should we train M6-1B (P2R), or a smaller model for longer.

2. The pretraining dataset is very small compared to the scale of the network. Practically speaking, training a multi-billion parameter model requires more data than BookCorpus and English Wikipedia, let alone a multi-trillion parameter model. The experimental results would be more convincing if the authors used larger datasets.

3. As the authors mentioned, we do not have results of finetuning the 10-trillion parameter model on downstream tasks. We don't know how impactful the proposed method can be.

4. The sharing-delinking paradigm is very similar to the sharing and unsharing method proposed in a previous ICLR submission: https://openreview.net/forum?id=jz7tDvX6XYR . The authors should discuss this work, even if it's a resubmission by the same authors, since the claims and experiments are different.

**Summary Of The Paper:**

The authors propose to train very large neural language models via a "Sharing-Delinking" paradigm. The proposed method first trains a model with weights shared across layers. In this way, the model appears to be smaller and it can fit into fewer GPUs. At some point, the authors delink the weights, and continue training the model in the conventional way.

The authors also propose a granular CPU offloading mechanism to save CPU memory.

**Summary Of The Review:**

It is very impressive to be able to train a 10-trillion parameter model. However, the benefit of training such a large model is unknown, since critical evaluations are missing. Overall, it appears to me that the experiments on very large models are not ready, given the lack of a reasonably sized dataset or downstream evaluations. Thus, it is not clear how impactful the proposed method can be.

---

> ### Author Response · Authors · 2021-11-17
> **Authors' response**
>
> Thank you very much for your valuable reviews. Here are our response to the concerns raised above:
> 1. About the comparison between M6-10T and M6-T: We believe that for models of different scales, sample-basis evaluation should be the most proper one. It is almost impossible to find that a 10 times larger model converges faster given a limited budget. We try to point out that "greener" means you don't need 10 times of budget even if you increase the model scale by 10 times.
> 2. About the pretraining dataset: We demonstrate the effectiveness of the paradigm and thus we use a basic dataset for better comparison with the baselines. For training extreme-scale models, we use large corpus like M6-corpus with hundreds of milllions of samples.
> 3. About the downstream performance of M6-10T: Due to the limitation of computation resources and time budget, currently we could not provide downstream results and thus we evaluated the model by log perplexity. According to the experience of pretraining on smaller-scale models, in general we believe the pretraining PPL and downstream performance are in positive correlation. In the future, we will attempt to add more downstream experiments for the extreme-scale model.
> 4. About the missing reference: This work was posed on arxiv after our submission, and we did not notice this paper among the rejected ICLR papers last year. Even so, in our submisson what we concern are different and we implement the mechanism on models with different architectures and different sizes.

---

### Official Review · Reviewer_3GYN · 2021-11-06

**Correctness:** 4
**Technical Novelty And Significance:** 3
**Empirical Novelty And Significance:** 3
**Recommendation:** 5
**Confidence:** 4

**Main Review:**

**Pros:**
1. The paper focuses on training extremely large models with limited computational resources, which is of great empirical importance.

2. The proposed P2R method is easy to implement and can be adapted to different models (as long as there is a decent number of repeated layers, whose weights can be shared).

3. The efficacy of P2R is empirically verified.

**Cons:**
1. *Insufficient related work discussion on fast training methods*. There is a large volume of works on fast training of deep learning models, and the notion of "weight sharing" was also explored previously. For instance, "Trellis networks for sequence modeling", "Speeding up Deep Model Training by Sharing Weights and Then Unsharing" show that sharing weights leads to faster convergence. Other works that train a smaller model first and then grow the model into a large one can also sometimes be recast as sharing and delinking (for example, "Net2Net: Accelerating Learning via Knowledge Transfer"; "Multi-level Residual Networks from Dynamical Systems View"; "Efficient Training of BERT by Progressively Stacking").

2. *Need a more comprehensive empirical evaluation*. Given that the main contribution of this paper is to propose a novel empirical training method, it needs more thorough evaluations. In particular, I think it would be good to have experiments covering: a) various models. Only M6 models are trained in this paper. The scale of M6-10T is promising, but it would be good to include other models which also have repeated layers. b) various downstream tasks. Evaluations only on WikiText-103 and Gigaword are somewhat limited. c) various domains. There are models commonly used in other domains (e.g., image classification) that have repeated layers. Does the proposed method work for them?

3. *Clarifications needed for Section 4.3*. In the training efficiency, the M6-10T with P2R is trained for around 10 days and another M6-10T is trained without P2R for around 3 days for comparison. Why is P2R training time longer than without P2R?

Figure 4(a) plots the Log Perplexity v.s. clock time and the conclusion is "P2R can outperform the one trained from scratch by a large margin". I feel this claim is not fully supported as the M6-10T does not converge yet. Give both more time, will they converge to a similar performance?

Figure 4(b) compares M6-10T P2R v.s. M6-T on sample-basis. Why the baseline is changed here and how does M6-10T P2R compare to M6-10 Real on a sample basis? Further, both do not seem to converge given 10M samples. How will the curve look like when giving more training samples (e.g., 20M)?

It also confuses me a little as the M6-10T P2R curves in 4(a) and 4(b) do not look the same - I'm confused because I think they should only differ by some scaling in the x-axis (changing from training time to samples trained)?

**Questions:**

1. For Table 1, the speed difference between Pseudo and Real is huge. What leads to the acceleration? Is that primarily coming from the GPUs' communication overhead? It seems that both Pseudo and Real are trained on 48 GPU devices according to the description, then they should have similar communication overhead?

2. For Figure 4, when does the "delink" happen? How frequently does P2R check for switching, and is that time included in Figure 4(a)?

3. After delink, the "Real Giant" will require the same amount of computational resources as training the model in the standard way? Is the computational saving then only appears in the "Pseudo Giant" stage? If so, what is the typical training time spent in the "Pseudo" and "Real" stages, respectively?

**Minor Comments:**

1. In the second paragraph of Section 3.2.1, it says "the amount of gradients becomes 2/L of the original one". Why not 1/L?

2. In the description of Table 1, $l$ should be $L$.

3. It's probably better to use a larger fontsize for the numbers and legends in Figure 2 and 4.

**Summary Of The Paper:**

This paper illustrates the training difficulty of extreme-scale models (with a special focus on memory requirement) and provides a simple but effective solution called "Pseudo-to-Real", where in the "pseudo" stage the weights of the repeated layers are shared, and in the "real" stage the weights are delinked. Experiments are conducted to demonstrate the efficacy of the proposed method.

**Summary Of The Review:**

This paper proposes an empirical method for fast training of models with repeated layers. The M6-10T experiments warrant scalability. However, I believe a more comprehensive set of experiments is needed to demonstrate the efficacy of the proposed training paradigm. Please see detailed comments in the main review.

---

> ### Author Response · Authors · 2021-11-17
> **Authors' response**
>
> Thanks very much for your informative review. Below is our response to the challenges and questions.
> 1. About missing references: we admit that we miss some of the important papers, such as "Efficient training BERT" that it stacks by using learned weights (which is essentially different from our method) and we will add them in our references in the future version. However, some works like "Net2Net" or "Multi-level" are not highly related to our paper, though they are concerned with efficient training.  Our paradigm is similar to the one in "Speeding up", but we focus on training different models and we find it extremely useful only when the model requires high memory footprints. We also extend it to training large MoE models, which should be another contribution. Furthermore, note that this work was actually posted on arxiv after our submission, and we believe that not citing it should not be a weakness.
> 2. About adding more experimental results: We appreciate this issue and have updated a new version of the paper, which includes experiments on more types of downstream tasks (such as GLUE benchmark). Please refer to section 3.3 in the updated version for more details. Meanwhile, we conduct more comprehensive comparisons between the Pesudo, P2R and Real models under the settings of training both for a long time and for only a limited time budget, respectively. The result supports the necessity of adding a P2R phase following the Pesudo phase, as well as the equality on wall-time downstream performance between P2R and Real models. In the future, we will try to explore the paradigm on more model architectures and downstream tasks.
> 3. About Sec 4.3: P2R is designed for training extremely large models with time and energy constraints. In our case, we cannot afford to train M6-10T with Real training from scratch. Instead, we provide the evidence in relatively small model. Also, we find that M6-10T with Real training only suffered from slow convergence at the 3rd day and it is reasonable to believe that there will still be a large performance gap between the models at the 10th day.
> 4. About the acceleration: We find that this strategy is highly effective for models requiring high memory footprint, namely large models. These model weights usually can take up much of the memory and thus squeeze the throughput of data. We try to imitate the extreme-scale training by implementing experiments on relatively small models. The models mentioned in Sec. 3 are large enough to take up most GPU memory. We find that the training speeds of Pseudo model and Real model are quite different in this scenario.
> 5. About the switching mechanism: Our switching method does not increase computation significantly. There is no need to stop Pseudo training for checking the best switching points. It is possible to continue Pseudo training while we test P2R with the saved checkpoints.
> 6. About time saving: The time saving mostly happens at the Pseudo stage. For large models that reuiqre high memory footprint, their training speed between Pseudo and Real training can be quite different, as mentioned above.
> 7. About the size of gradients: As we have a large computation graph but much fewer parameters, the size of weight parameters should be 1/L, but the size of gradients should be L. However, we replace the "add" operation with "accumulate", and thus we only need 2/L for accumulation each time.

---

### Official Review · Reviewer_gmHp · 2021-11-06

**Correctness:** 2
**Technical Novelty And Significance:** 3
**Empirical Novelty And Significance:** 2
**Recommendation:** 3
**Confidence:** 4

**Main Review:**

Strengths:
* Overall, the paper is well-written and the contributions of authors are clearly explained.
* To my knowledge, this is the first work that leverages both expert sparsity and parameter offloading to train a model of unprecedented scale, which is an important engineering achievement if it was indeed executed succesfully.

Weaknesses:
* The overall claim of pretaining a 10-trillion parameter model in 10 days is a bit misleading: first, the stopping criterion for pretraining is unclear, that is, it's not entirely clear whether the model has converged after just 15k steps and other models of this scale took significantly more steps for pretraining with even larger batches. Second, there is no downstream evaluation for M6-10T or comparison with other models of comparable scale: it might have been possible to reimplement some of the prior work as baselines.
* Although pretraining with parameter sharing may indeed be more memory-efficient, the (seemingly) inherent disadvantage of P2R is that in the second stage of training, that efficiency is lost due to the unlinking. As a result, the users of this methodology will still be forced to train the model of full size at some point, and the gains of P2R are not fully quantified. For instance, why not just pretrain the model with parameter sharing for the entire duration of the experiment, as in the original ALBERT paper? Moreover, even if the wall-time convergence of P2R is much faster, how does Pseudo pretraining compare with Real in terms of iteration convergence?
* The pretraining objectives, such as text denoising and image-based text denoising, are not described in detail. Authors should provide the code of the objectives, describe it in the paper or at least give the exact reference to prior work, because there exist several different objectives which fall into this category. For instance, it is unclear how the resulting model is applied to the language modeling and summarization tasks, because in some definitions of denoising pretraining, the model actually "sees" the entire input sequence and thus does not learn to generate it in an autoregressive manner.
* In Section 3.2.3, authors note that in order to determine the moment of pseudo-to-real switching, they need to repeatedly attempt to swiitch at different training steps and measure the difference in loss between continuing to train with shared parameters and training with unlinked ones. This strategy seems to highly increase the computation load of the entire procedure, unless in practice the intervals between attempts and the evaluation period are very conservative: the exact values for the main experiment as well as the number of "unsuccesful" attempts are not given in the paper, and I believe that this issue needs to be discussed in more detail.
* Looking at the results in Table 1, it was quite surprising to observe a difference in throughputs of Pseudo and Real which is this significant. Although the parameters are shared across layers, the computational requirements of both forward and backward passes should be approximately the same. While there might be some additional inefficiencies connected with the use of pipeline parallelism, the work does not discuss or analyze this difference in performance; in the original ALBERT paper, authors have found training with shared parameters to be only 1.7 times faster than without sharing. I believe that the comparison should be without offloading, as it is an orthogonal factor that needs to be evaluated separately.
* The granular offloading mechanism also needs to be compared both with ZeRO-Offload [1] and L2L [2] in terms of its efficiency. As of now, it is not quite clear whether the bottleneck outlined by authors is indeed a problem: in [1], the authors attempt to address the communication bottleneck by overlapping it with computation and asynchronous optimization.
* Lastly, using idle GPU memory for storing extra layers means less training examples processed in a microbatch for pipelining or gradient accumulation, which may also reduce the efficiency of distributed training. Currently, there is no discussion of this tradeoff, and this question might be important in practice, since currently the most popular approach is to train models of this scale on extremely large batches.

Questions and typos:
* In Section 3.2.1, you say that "the amount of gradient [when sharing parameters] becomes $2/L$ of the original one"; if there are L times fewer parameters, shouldn't that be just $1/L$, the same factor which you arrived at for weights and optimizer states?
* For the M6-10T model, what datasets and pretraining objectives did you use? Is it the same data and tasks as in Section 3.3.1?
* Small nitpick: the methods offered as a part of the DeepSpeed framework are grouped under the name of ZeRO, not ZERO.

[1] ZeRO-Offload: Democratizing Billion-Scale Model Training. Jie Ren, Samyam Rajbhandari, Reza Yazdani Aminabadi, Olatunji Ruwase, Shuangyan Yang, Minjia Zhang, Dong Li, Yuxiong He
[2] Training Large Neural Networks with Constant Memory using a New Execution Algorithm. Bharadwaj Pudipeddi, Maral Mesmakhosroshahi, Jinwen Xi, Sujeeth Bharadwaj

**Summary Of The Paper:**

This work studies the problem of efficient pretraining of large-scale models for language and vision representations, namely the issue of significant memory requirements for models with billions to trillions of parameters. Authors propose two modifications: first, to reduce the memory load and improve convergence at the initial stage of training, they suggest to train a multilayer model with shared parameters and then unshare them. Second, to maximize GPU utilization with offloading, authors develop a method for granular CPU offloading, which keeps larger chunks of the model in GPU memory. When combined, the proposed methods allow the authors to train a 10 trillion parameter model on 512 GPUs.

**Summary Of The Review:**

This work proposes two methods for improving the efficiency of large-scale deep learning. While the methods are conceptually very simple and thus compelling, the paper might benefit from a more thorough validation of claims about these methods, both in terms of compaing with prior work and more detailed empirical evaluation.

---

> ### Author Response · Authors · 2021-11-17
> **Authors' response**
>
> Thanks for your helpful reviews. Below we address your concerns:
> 1. About the misleading claim: We determine the stopping criterion by evaluating the log perplexity. Admittedly, it was not available for us to provide downstream results due to resource limitation. Yet we find that P2R is highly necessary for training a giant model with a limited budget. The Pseudo stage helps the model converge fast to a low log perplexity, and the Real stage helps the model reach the best performance. We show that P2R can be competitive with Real training in the experiments for small-scale models.
> 2. About the performance comparison between Pesudo, P2R and Real: We have updated the paper with more experimental details about the downstream performances of Pesudo, P2R and Real models. Please refer to section 3.3.1 and 3.3.2 of the updated version, where we provide the comparisons under the settings of training for a long time and for a limited time budget, respectively. On both settings, the experimental results indicate that the Pseudo model significantly sacrifices downstream performance in return of much faster speed, for which a following P2R phase is necessary to overcome this issue.
> 3. About the objectives: we follow M6 and apply the same architecture and training objectives, and thus we did not present the details of the objective functions, which are not our focus in this paper. We will point out the related work and add the details in our appendix in the next version. As to the questions about the objectives, for plain text data, we simply use text denoising for learning to recover masked texts. However, we put them in a same encoder and apply a masking strategy so that the model is able to process the masked input information bidirectionally and learns next-token prediction in an autoregressive manner to recover the original text. As to the downstream transfer, as the model learned language modeling already, we can directly use it for zero-shot evaluation of its capability of language modeling by calculating the PPL. For summarization, we finetune the model on the downstream dataset. Similar to BART or T5, in downstream finetuning, the model encodes the source text and decodes the target.
> 4. About the switching mechanism: our switching method does not increase computation significantly. There is no need to stop Pseudo training for checking the best switching points. It is possible to continue Pseudo training while we test P2R with the saved checkpoints.
> 5. About the size of gradients: As we have a large computation graph but much fewer parameters, the size of weight parameters should be 1/L, but the size of gradients should be L. However, we replace the "add" operation with "accumulate", and thus we only need 2/L for accumulation each time.
> 6. About Granular offloading: Besides the issue of efficiency, another concern of our Granular offloading is the size of memory. For training extremely large models, the memory can be a bound for ZeRO-offload, while the granular offloading can alleviate the problem. In our practice of training the 10T model, it is impossible to offload the whole model to the CPU memory due to the memory limitation. We will add more quantitative analysis about its efficiency in comparison with the original offloading strategy for better understanding.

---

### Decision · Program_Chairs · 2022-01-20

**Decision:**

Reject

**Comment:**

This paper proposed a weight sharing method to speed up the pretraining of large language models. Basically, during the training, it first share weights across all the layers with the same architecture, and then untie the shared weights at some point later. The main advantage of weight sharing is that it can reduce the memory load. Our reviewers have many concerns on this work. The method is not well motivated or explained, and many experimental details are missing. In particular, there is no downstream task result presented for the so-called 10T parameter model. The claim highlighted in the title remains unsupported. In addition, one of our reviewers pointed out that the proposed method is fairly similar to the method in a previous ICLR submission: https://openreview.net/forum?id=jz7tDvX6XYR.